# Lag in Hydrologic Recovery Following Extreme Meteorological Drought Events: Implications for Ecological Water Requirements

**Qiang Liu [1,\*], Xiaojing Ma [1,2], Sirui Yan [1], Liqiao Liang [3], Jihua Pan [2] and Junlong Zhang [4]**

[1] State Key Laboratory of Water Environment Simulation, School of Environment, Beijing Normal University, Beijing 100875, China; maxiaojing228@163.com (X.M.); yansirui@mail.bnu.edu.cn (S.Y.)

[2] College of Geography and Tourism, Qufu Normal University, Rizhao, Shandong 276826, China; pjh78119@163.com

[3] Key Laboratory of Tibetan Environment Changes and Land Surface Processes, Institute of Tibetan Plateau Research, Chinese Academy of Sciences, Beijing 100101, China; liangliqiao@itpcas.ac.cn

[4] College of Geography and Environment, Shandong Normal University, Jinan, Shandong 250358, China; junlongzhangcq@hotmail.com

\* Correspondence: qiang.liu@bnu.edu.cn

**Abstract:** Hydrological regimes, being strongly impacted by climate change, play a vital role in maintaining the integrity of aquatic river habitats. We investigated lag in hydrologic recovery following extreme meteorological drought events, and we also discussed its implications in the assessment of ecological environment flow. We used monthly anomalies of three specific hydrometeorological variables (precipitation, streamflow, and baseflow) to identify drought, while we used the Chapman–Maxwell method (the CM filter) with recession constant calculated from Automatic Baseflow Identification Technique (ABIT) to separate baseflow. Results showed that: (i) Compared to the default recession parameter ($\alpha$ = 0.925), the CM filter with the ABIT estimate ($\alpha$ = 0.984) separated baseflow more accurately. (ii) Hydrological drought, resulting from meteorological drought, reflected the duration and intensity of meteorological drought; namely, longer meteorological drought periods resulted in longer hydrological drought periods. Interestingly, the time lag in streamflow and baseflow indicated that aquatic ecosystem habitat recovery also lagged behind meteorological drought. (iii) Assessing environmental flow by quantifying drought provided greater detail on hydrological regimes compared to abrupt changes, such as the increased hydrological periods and the different environment flows obtained. Taken together, our results indicated that the hydrological response in streamflow and baseflow (e.g., the time lag and the precipitation recovery rate ($P_r$)) played a vital role in the assessment of environmental flow.

**Keywords:** baseflow; extreme drought; recession parameter; Taoer River basin

## 1. Introduction

Baseflow is defined as being either flow-derived from subsurface flow or other delayed water sources, which sustains streamflow during dry periods [1,2], thus playing a vital role in perennially maintaining aquatic habitats [3,4]. Extreme meteorological events, particularly extreme meteorological drought events, dramatically influence hydrological regimes while also altering baseflow [5]. Alterations in hydrological regimes have led to the degradation of aquatic ecosystems [6]. In this context, there is a growing need to assess hydrological response characteristics and estimate environmental flow (e-flow) to maintain the integrity of aquatic ecosystems.

Over the past decade, greater than 170 hydrologic metrics have been developed to attempt to describe different components of flow regimes and capture ecologically relevant streamflow attributes [7]. Over time, several aspects of concern associated with e-flow have changed from the maintenance of minimum or suitable flow to hydrological regimes, including the magnitude, frequency, duration, timing, the rate-of-change of flow conditions, interannual variability, and the predictability of flow events [8,9]. Applying the natural flow regime concept, e-flow has allowed us to bridge our understanding of the connection between hydrology and ecology [10], such as quantifying hydrologic alterations (e.g., Richter et al., 1996 [11]) and flow–ecology relationships (e.g., King et al., 2016 [12]), and to construct holistic frameworks to manage rivers that have been altered by flow regimes [13]. Compared to the assumptions on "stationary" climatic and ecological processes being a "reference" condition (natural flow regime), "non-stationarity" climatic (as well as other environmental conditions) and ecological features present important challenges in understanding e-flow [13]. Although the effects of time-varying flow regimes are rare, understanding the ecological community response to dynamic flow regimes has engendered considerable attention [14]. For example, baseflow has been separated from streamflow to express its vital role in maintaining aquatic habitat connectivity (e.g., Beatty et al., 2010 [15]). Baseflow separation, and its response to extreme meteorological events, will help us better understand interactions between flow regimes and habitats [5,16,17], which will also help us to assess e-flow. Accordingly, the objectives of this study were: (i) to separate baseflow using the Chapman–Maxwell method (the CM filter) with recession constant estimated from the Automatic Baseflow Identification Technique (ABIT); (ii) to assess the response of hydrological drought characteristics to extreme meteorological drought; and (iii) to estimate the attribution of baseflow to e-flow in the Taoer River basin, China. This study explored the hydrological response to extreme meteorological drought and its implications in maintaining the integrity of aquatic ecosystems.

## 2. Materials and Methods

### 2.1. Study Site and Data

The Taoer River is in northeastern China, originating from the Greater Khingan mountain range, with a total drainage area of approximately 42,000 km$^2$. The basin above the Taonan station was selected as the study area, with a drainage area of approximately 36,000 km$^2$ (Figure 1). The Taoer River basin belongs to a mid-temperate continental monsoon climate, with an average annual precipitation of approximately 390 mm and a pan evaporation of approximately 1800 mm [18]. Given that the river flows through semiarid and semi-humid areas, the eco-environment is fragile in at least part of the basin [19]. Influenced by the combined effects of climate change and anthropogenic activities, several eco-environmental problems have predominated, including desertification, salinization, a decrease in wetland area, and an increased seasonality of rivers. For example, zero streamflow has been observed over a period of 13 years between 1990 and 2005 in the Taoer River. Zero streamflow at Taonan, a control station in the middle reach of the Taoer River, was reported in 2002, 2003, and 2004 [20]. Furthermore, the Momoge National Nature Reserve wetland has also shown signs of degradation due to water loss in natural hydrological regimes previously maintained by floodwater inflow from the Neijiang River and the Taoer River [21]. In this context, it is critical to understand alterations in flow regimes and their associated regulations on wetland structure and function [22].

Daily precipitation series between 1959 and 2016 from four local meteorological stations were provided by the National Climatic Center of the China Meteorological Administration (http://data.cma.cn/) to explore extreme climate events in the region. Daily streamflow data of Taonan hydrological station for the corresponding period were provided by the Jilin Hydrological and Water Resources Survey Bureau, Jilin Province, and used to investigate the hydrological response to an extreme drought event in the Taoer River basin, China.

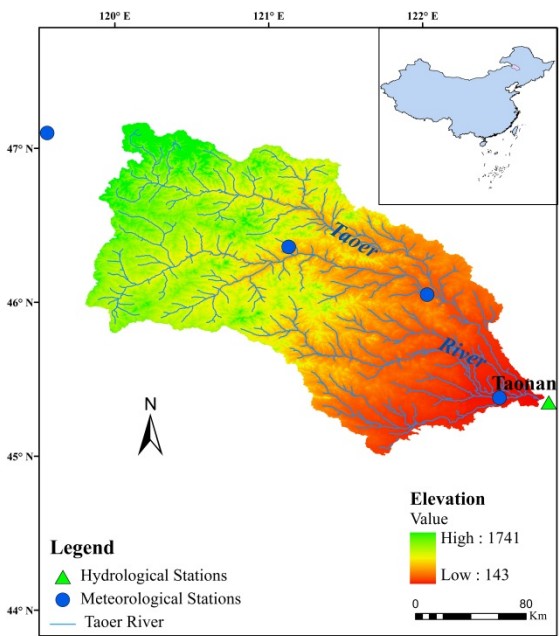

**Figure 1.** The location of the Taoer River basin, China. Meteorological stations (●) and the hydrological station (▲) are also shown.

*2.2. Methods*

2.2.1. Quantifying Drought

The definition of drought has been described as "inputs with the recognition that this meteorological definition may lead to a variety of responses depending on ecosystem hydrology" [23]. This definition provides a new perspective to distinguish between the status of meteorological, hydrological, agricultural, and socioeconomic hydrological conditions [24]. Considering its elusive, slowly developing characteristics, it is difficult to determine the onset and conclusion of drought with accuracy [25]. Many methods have been used to define drought, such as the Palmer Drought Severity Index (PDSI), the Standardized Precipitation Index (SPI), the Palmer Hydrological Drought Severity Index (PHDI), and the Surface Water Supply Index (SWSI) [25]. In this study, drought was quantified through the following steps [5]: (i) by calculating the annual precipitation anomaly, wherein results are smoothed within a three-year moving window to avoid individual wetter years that can be interspersed between extensive and pronounced dry periods and (ii) by determining the exact months for the onset and the conclusion of the dry period based on accumulated monthly precipitation anomalies. This study used a piecewise regression model for the cumulative monthly anomaly series to detect turning points in this series:

$$y = \begin{cases} \beta_0 + \beta_1 t + \varepsilon & t \leq \alpha \\ \beta_0 + \beta_1 t + \beta_2(t - \alpha) + \varepsilon & t > \alpha \end{cases} \tag{1}$$

where $t$ is the month and $y$ is the accumulated monthly precipitation anomaly; $b_0$, $b_1$, and $b_2$ are the regression coefficients; and $\alpha$ is the assumed turning point based on annual anomaly analysis. The range of the $\alpha$ value was set to 12 months prior to and following the start and end years that were determined through annual anomaly analysis. Linear least squares regression was used to estimate the three regression coefficients, and a *t*-test was applied to test whether or not $\beta_2$ equals zero. Drought duration was determined as the time difference between the start and end months. Drought severity was calculated as the accumulative precipitation anomaly during the drought event, while drought intensity was calculated as the ratio of drought severity over drought duration. In this way, hydrological drought can also be estimated using total streamflow and baseflow data.

### 2.2.2. Isolating Baseflow from Total Streamflow

Numerous methods have been developed to separate baseflow from total streamflow, which can be categorized as tracer-based and non-tracer-based methods [26]. Zhang et al. (2017) [16] evaluated four baseflow separation methods in eastern Australia (including the United Kingdom Institute of Hydrology (UKIH) method and three digital filtering methods: the Lyne–Hollick method, the Chapman–Maxwell method (the CM filter), and the Eckhardt method), and they reported that it is critical to obtain appropriate parameters before applying any digital filtering method. This study used the CM filter to partition streamflow into baseflow and quickflow, which is a new algorithm which has been applied to the digital filtering method [16]. The CM filter algorithm is determined as follows [16,27–29]:

$$Q_{b(i)} = \frac{\alpha}{2-\alpha} Q_{b(i-1)} + \frac{1-\alpha}{2-\alpha} Q_i \qquad (2)$$

where $Q$ is the total streamflow (mm/d); $i$ is the time step (daily); $\alpha$ is the filter parameter (recession constant, units of 1/day); and $Q_b$ is baseflow (mm/d). Due to the prevalence of this linear scenario within natural river systems [30], the theoretical equation to describe baseflow can be expressed as follows:

$$\frac{dQ}{dT} = \frac{1}{\kappa} Q \qquad (3)$$

where $T$ is the time step (daily), and $\kappa$ is the characteristic drainage time scale (days). The recession constant $\alpha$ can be inferred as follows:

$$\alpha = e^{-\frac{1}{\kappa}} \qquad (4)$$

In this study, the recession analysis method proposed by Brutsaert and Nieber [31] (denoted as the BN77 method) was employed to estimate the catchment recession constant $\alpha$. ABIT was used to calculate the catchment recession constant $\kappa$ [32].

### 2.2.3. Temporal Trends in Streamflow and Assessment of Ecological Water Requirement

To assess the temporal trends of the streamflow data series, ordinary linear regression and the Mann–Kendall (M–K) test were used. Ordinary linear regression, which here tested against the hypothesis of null slope by means of a two-tailed *T*-test at a confidence level of 95% [33], is a common method of statistical diagnosis in modern hydrometeorological analysis. The slope of linear fitted model and $R^2$ were got from the ordinary linear regression. The M–K test has been widely used to test trends in hydrological and climatological time series [34,35]. The M-K rank statistics ($S_u$) were calculated, and two curves of the M-K rank statistics $C_1$ and $C_2$ used to test the abrupt changes. If $C_1$ exceeds the confidence line, it means that there is a significant upward or downward trend in series. Moreover, if the intersection point of the $C_1$ and $C_2$ is between the two confidence lines, we can consider that abrupt climate change takes place at that point; otherwise, we should diagnose it combined with other methods (e.g., the Yamamoto method) [35]. The M–K method was used to obtain the timing of abrupt changes in streamflow, combined with the Tenant method to define the ecological water requirement in different periods.

The Tenant method, the most popular historical streamflow method, has been widely applied to assess the environment flow [36,37]. The Tennant method is based on a fixed percentage of mean annual flow, linking the rate of change in river hydraulic parameters (mainly widths, depths, and velocities) at flows with habitat [36,38]. Usually, the flow between 60% and 30% of the mean annual flow (MAF) are acceptable levels to maintain good fish habitat, and a minimum 10% of the MAF was regarded as being necessary for maintenance of aquatic ecosystems [36,37]. In this study, 10%, 30%, and 60% of MAF was calculated as the minimum ($E_{wm}$), basic ($E_{wb}$), and suitable ($E_{ws}$) environmental flow (e-flow).

## 3. Results

### 3.1. Baseflow Separation and Its Validation

The time scale $\kappa$ and the recession constant $\alpha$ were obtained using ABIT (Figure 2), wherein the former and the latter for the Taoer River were 61.3 days and 0.984, respectively. The recession constant $\alpha$ was larger than the default recession constant (0.925).

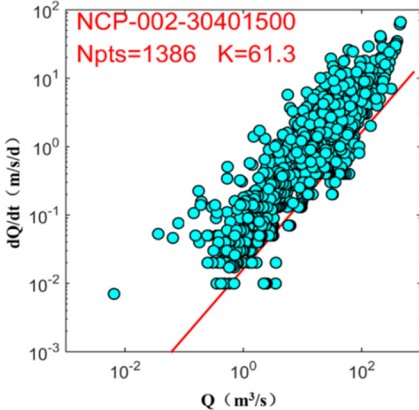

**Figure 2.** Estimation of the recession constant (plot of log (*dQ/dt*) and log (*Q*) for the Taoer River basin) using the Automatic Baseflow Identification Technique (ABIT). The red line denotes the lower 5% envelope with a slope of 0.984, while the estimate of the characteristic drainage time scale was $\kappa$ = 61.3 days.

In order to validate the recession constant $\alpha$, baseflow was calculated using both the recession constant (0.984) and its default value (0.925) (Table 1). Baseflow accounted for 37.73% and 48.24% of streamflow when recession constant $\alpha$ was 0.984 and 0.925, respectively. As 0.925 in the default method is defined from six catchments in Germany by Nathan and McMahon [39], the CM filter with calculated recession constant reduces the uncertainties in specific catchment and provide suitable result in separating baseflow.

**Table 1.** Attribution analysis of baseflow using the Chapman–Maxwell method (the CM filter) and the two-terminal mixed water source segmentation model.

| Method | Recession Constant ($\alpha$) | | Attribution (%) |
|:---:|:---:|:---:|:---:|
| **The CM filter method (recession constant)** | Default | 0.925 | 48.25 |
| | ABIT | 0.984 | 37.73 |

### 3.2. Meteorological and Hydrological Drought Characteristics

Two extreme meteorological drought events were detected according to precipitation anomaly characteristics (Figure 3). The duration of these two meteorological drought events was 93 months (June 1975–March 1983) and 137 months (April 1999–September 2010), respectively.

The severity and intensity of the meteorological drought events as shown by the three hydrometeorological variables are shown in Table 2. Meteorological drought severity and intensity were greater during the second drought period with values of −16.84% and −0.12%, respectively. Compared to meteorological drought, hydrological drought severity and intensities in streamflow and baseflow were greater with average values of −45.75% and −0.38% (first period) and −82.61% and −0.51% (second period), respectively. The time lag between the end of meteorological drought and the end of hydrological drought showed that baseflow recovery exhibited a longer lag time (28 and 30 months for the first and second drought periods, respectively) compared to streamflow recovery (27 and 22 months for the first and second drought periods, respectively). The recovery rate of precipitation

exhibited a consistent trend along with time lags with values of −14.39% and −40.35% for baseflow and −12.09% and −9.47% for streamflow during the first and second drought periods, respectively.

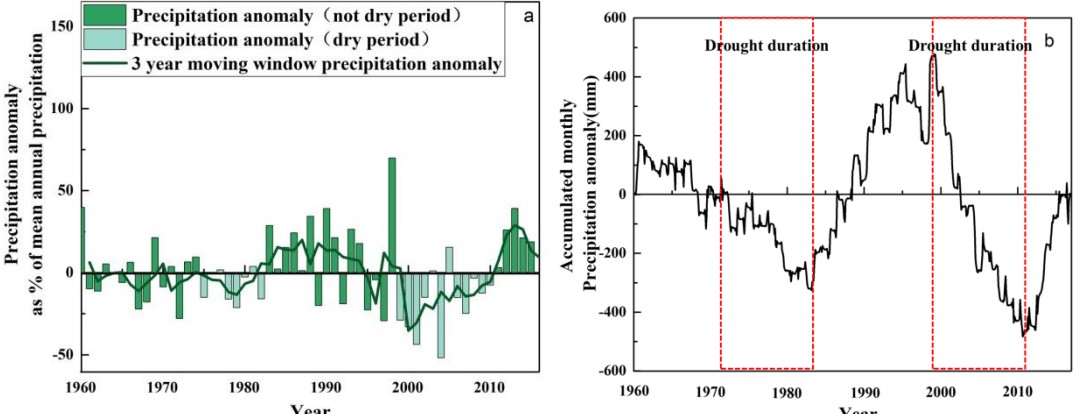

**Figure 3.** Assessment of meteorological drought events using precipitation anomalies. (**a**) start and end years of meteorological drought; (**b**) exact month of the start and end of the drought period.

**Table 2.** Meteorological (precipitation, *P*) and hydrological (streamflow, *Q*; baseflow, $Q_b$) characteristics of the Taoer River basin, the start and end times, duration, intensity, severity, time lags, and the precipitation recovery rate ($P_r$, calculated as the ratio of the accumulated relative precipitation anomaly between the end of meteorological drought and the end of hydrological drought over the time lag, % per month).

| Variables | Periods | Start Time | End Time | Duration | Intensity | Severity | Time Lags | $P_r$ |
|---|---|---|---|---|---|---|---|---|
| | | (Year.month) | (Year.month) | (Month) | (%) | (%) | (Month) | (%) |
| *P* | | 1975.6 | 1983.3 | 93 | −0.073 | −6.80 | − | − |
| *Q* | First | 1975.6 | 1985.7 | 121 | −0.36 | −44.16 | 27 | −12.09 |
| $Q_b$ | | 1975.7 | 1985.8 | 121 | −0.39 | −47.34 | 28 | −14.39 |
| *P* | | 1999.4 | 2010.9 | 137 | −0.12 | −16.84 | - | - |
| *Q* | Second | 1999.6 | 2012.8 | 158 | −0.52 | −81.70 | 22 | −9.47 |
| $Q_b$ | | 1999.7 | 2013.4 | 165 | −0.51 | −83.43 | 30 | −40.35 |

### 3.3. Assessment of Ecological Environmental Flow (e-flow)

Abrupt changes in combination with the ecological water requirement method have been widely used to estimate e-flow. According to M-K method and Yamamoto method, three abrupt changes have been detected in annual streamflow in the Taoer River (Figure 4), which separated annual streamflow into four hydrological periods.

The minimum ecological water requirement ($E_{wm}$) ranged from 0.64 (drought periods) to 1.55 (wet periods) $\times 10^8$ m$^3$ during these four hydrological periods (Table 3). Combined with traditional ecological e-flow assessments, we obtained greater detail using the drought quantification method (as shown in Table 3). In particular, the drought quantification method can separate hydrological periods on a monthly scale. Furthermore, more detail on e-flow can be deduced from different hydrological periods, such as $E_{wm}$, which ranged from 0.17 to 2.0 $\times 10^8$ m$^3$ with an average value of 1.01 $\times 10^8$ m$^3$, the basic ecological water requirement ($E_{bs}$), and the suitable ecological water requirement ($E_{ws}$), which ranged from 0.50 to 5.99 and from 0.99 to 11.99 $\times 10^8$ m$^3$, respectively.

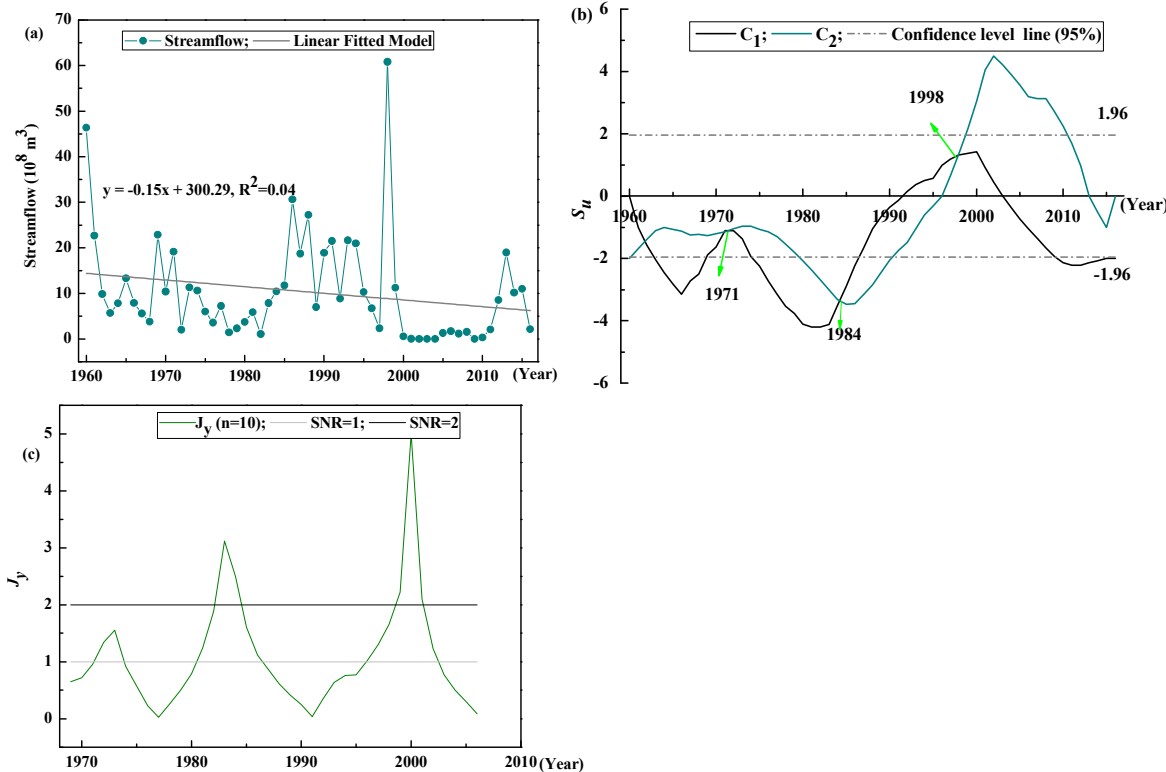

**Figure 4.** Temporal trends (**a**) and abrupt changes (**b, c**) in streamflow for the Taoer River. The timing of abrupt changes detected by Mann-Kendall method (**b**) and Yamamoto method (**c**) also showed in Figure 4b. $S_u$ is the M–K rank statistics, $C_1$ and $C_2$ are two curves of the Mann-Kendall rank statistics. The confidence level (95%) is also shown in the figure. The green arrow pointed the intersection point of the $C_1$ and $C_2$ to show the abrupt changes. For the Yamamoto method, $J_y$ is the signal-to-noise ratio; SNR = 1, at 95% significance level; SNR = 2, at 99% confidence level.

**Table 3.** The ecological water requirement for minimum ($E_{wm}$), basic ($E_{wb}$), and suitable ($E_{ws}$) environmental flow (e-flow) during different hydrological periods.

| Method | Period | Mean $Q$ | $E_{wm}$ (10%) | $E_{wb}$ (30%) | $E_{ws}$ (60%) |
|---|---|---|---|---|---|
| | (Year.month) | (10$^8$ m$^3$) | (10$^8$ m$^3$) | (10$^8$ m$^3$) | (10$^8$ m$^3$) |
| Hydrological drought quantification | 1960.01–1975.05 | 13.27 | 1.33 | 3.98 | 7.96 |
| | 1975.06–1985.07 | 4.96 | 0.50 | 1.49 | 2.97 |
| | 1985.08–1999.05 | 19.98 | 2.00 | 5.99 | 11.99 |
| | 1999.06–2012.08 | 1.65 | 0.17 | 0.50 | 0.99 |
| | 2012.09–2016.12 | 10.59 | 1.06 | 3.18 | 6.35 |
| Abrupt changes | 1960–1970 | 14.22 | 1.42 | 4.27 | 8.53 |
| | 1971–1983 | 6.35 | 0.64 | 1.91 | 3.81 |
| | 1984–1997 | 15.52 | 1.55 | 4.66 | 9.31 |
| | 1998–2016 | 6.95 | 0.69 | 2.08 | 4.17 |

## 4. Discussion

### 4.1. The Response of Streamflow and Baseflow to Extreme Meteorological Drought

A decreasing trend in streamflow has resulted in the drying up in the lower reaches of the Taoer River and has affected the integrity of the surrounding wetland (e.g., Li et al., 2013 [19]; Jiang et al., 2016 [21]). By analyzing anomalies of various hydrometeorological time series, we were able to determine two meteorological drought events in the Taoer River basin. The precipitation deficit for 1975–1983 and 1999–2010 could help to explain this decreasing trend in streamflow (Table 2). In fact,

meteorological drought in the Taoer River basin was also detected by other drought assessment methods (e.g., Song et al., 2015 [40]). Influenced by catchment characteristics, the duration of hydrological drought typically lasted longer than meteorological drought [5,41]. These results highlight the importance of analyzing drought in different catchments, which are relevant to ecological and hydrological impacts, particularly for aquatic ecosystems [14]. The time lag in hydrological recovery from meteorological drought (Table 2) also confirmed that the time required for the recovery of aquatic ecosystems from meteorological and hydrological droughts is extensive. As reported by Yang et al. (2017) [5], baseflow recovery generally exhibits a longer time lag compared to streamflow recovery. The reason for this mainly lies in the fact that baseflow recovery usually requires the catchment water storage to exceed a certain threshold, which is governed by multiple climatic factors and catchment properties [5,42]. As showed in Table 2, drought severity and intensity were greater (−47.37% and −0.39% in the first period and −83.43% and −0.51% in the second period, respectively), and baseflow recovery correspondingly exhibited a longer time (28 months for the first period and 30 months for the second period).

*4.2. Assessment Implications for Environmental Flow (e-flow)*

Considering that temporal variability in streamflow provides a range of ecosystem processes and habitat requirements that sustain high native diversity, e-flow, which is rooted in the natural flow regime paradigm, has been assessed and applied to restore a range of flow regimes [10,13]. However, flow regimes differ across different river types and climates and can also shift temporally through the impact of climate change, land-use type, or flow management practices [14]. As reported by Poff [13], researchers have become more attentive to the effect of time variation on flow regimes. As shown in Figure 3 and Table 1, it is difficult to define periods outlining natural flow regimes, and use this information to link flow regimes with ecological structure and function. Explanations for linkages between flow regimes, biota, and ecosystem processes will help define e-flow while also helping to protect and restore aquatic ecosystems [14]. Therefore, the challenges in determining e-flow involve transitions from relatively static, regime-based hydro-ecological characteristics to those that are more dynamic as well as time-varying flow characteristics that can be linked mechanistically to a wide range of ecological performance metrics applicable over a range of spatial and temporal scales [13]. Catchment characteristics also control streamflow response to climatic change (e.g., duration and the time lag of hydrological drought). For example, Yang et al. (2017) [5] argued that a nonproportional reduction between hydrological variables and precipitation data has caused incertitude in the assumption of catchment stability over long periods, which is implicitly embedded into many hydro-climatological models (e.g., the Budyko model); therefore, this requires additional research. Furthermore, groundwater plays a vital role in maintaining environmental water requirements for some special ecosystems, such as surface flow-dependent ecosystems [3,43,44]. In order to maintain a healthy ecosystem, de Graaf et al. (2019) [45] linked a decrease in the level of groundwater flow resulting from groundwater pumping to a decrease in streamflow globally, while also estimating the location and time of environmentally critical streamflow. Therefore, alterations in groundwater storage will affect time lag and precipitation recovery rates.

## 5. Conclusions

Baseflow regimes play a vital role in sustaining streamflow and maintaining integrity of aquatic ecosystems. This study assessed baseflow separated from streamflow as well as multiple aspects of meteorological and hydrological drought in the Taoer River basin, China. Two meteorological and relative hydrological drought events were tested using precipitation and streamflow anomalies. Results showed that the duration of hydrological drought was longer than meteorological drought, and the intensity of meteorological drought was significantly amplified in hydrological drought.

We determined that the time-lag and precipitation recovery rate data of the rainfall–runoff relationship were non-stationarity in nature over prolonged periods of drought, which must be

incorporated into e-flow assessments. Furthermore, dynamic, time-varying flow characteristics must also be incorporated into e-flow frameworks and reflect the effects of catchment characteristics in regulating flow regimes. All of these results should help address climate changes in regulating e-flow to maintain integrity of aquatic ecosystem.

**Author Contributions:** Conceptualization, Q.L.; Methodology, J.Z. and X.M.; Validation, Q.L. and L.L.; Formal analysis, Q.L. and J.P.; Investigation, X.M. and S.Y.; Resources, X.M. and S.Y.; Writing—original draft preparation, Q.L.; Writing—review and editing, Q.L. and L.L.; Project administration, Q.L. All authors have read and agreed to the published version of the manuscript.

**Funding:** This study was supported by the National Key R&D Program of China (No. 2016YFC0500402, 2017YFC0404505), the Major Science and Technology Program for Water Pollution Control and Treatment (No. 2018ZX07110001), the National Natural Science Foundation of China (No. 51579008), and Beijing Municipal Science and Technology Project (No. 217300011).

**Acknowledgments:** The authors of this study would like to thank the National Climatic Center of the China Meteorological Administration for providing the meteorological data. The authors would also like to thank the Jilin Hydrological and Water Resources Survey Bureau, Jilin Province, for providing the hydrological data.

**Conflicts of Interest:** The authors declare no conflict of interest.

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
