# Peer review of "Lag in Hydrologic Recovery Following Extreme Meteorological Drought Events: Implications for Ecological Water Requirements"

_water, doi:10.3390/w12030837_

Round 1

Reviewer 1 Report

Line 70-71: Authors pointed out that “ with an average annual precipitation of approximately 390 m and a pan evaporation of approximately 1800 mm.” Please check the annual precipitation again. Is it 390 m? Line 177: The result of baseflow by using default value (0.925) is larger (48.24%). It is necessary to explain this characteristic. Is this method of baseflow separation not suitable in this study area? Why? Line 183: In Fig. 3, It is better to do trend analysis for precipitation or drought. Is the precipitation trend positively increasing or negatively decreasing? Line 201: In Table 2, It is necessary to explain how to determine the start time and end time of data period (1975.6-1985.7; 1975.7-1985.8; 1999.6-2012.8; and 1999.7-2013.4) during the drought events. Different time period could get different results. Line 235-238: The baseflow recovery in this study area generally exhibits a longer time lag. How do the climatic factors and catchment properties in your study area cause the result of longer time lag? It is necessary to explain in more detail.

Author Response

Line 70-71: Authors pointed out that “ with an average annual precipitation of approximately 390 m and a pan evaporation of approximately 1800 mm.” Please check the annual precipitation again. Is it 390 m?

Response: This mistake has been corrected.

Line 177: The result of baseflow by using default value (0.925) is larger (48.24%). It is necessary to explain this characteristic. Is this method of baseflow separation not suitable in this study area? Why?

Response: Because of the 0.925 in the default method is defined from 6 catchments in Germany by Nathan and McMahon (1990). This is more suitable for those catchment which catchment attributes are close to his study. In this study, to reduce the uncertainties we used the revised/improved baseflow separation method improved by Zhang et al. (2017). The recession analysis is proposed by Cheng et al. (2016), and this automatic procedure can provide the specific recession constant from each catchment (no constant 0.925).

Nathan, R., McMahon, T., 1990. Evaluation of automated techniques for base flow and recession analyses. Water Resour. Res. 26 (7), 1465–1473. http://dx.doi.org/10.1029/WR026i007p01465.

Zhang, J. L., Zhang, Y. Q., Song, J. X., & Cheng, L. (2017). Evaluating relative merits of four baseflow separation methods in Eastern Australia. Journal of Hydrology, 549, 252-263. doi: 10.1016/j.jhydrol.2017.04.004

Cheng, L.; Zhang, L.; Brutsaert, W. Automated Selection of Pure Base Flows from Regular Daily Streamflow Data: Objective Algorithm. J. Hydrol Eng. 2016, 21(11), 06016008. http://dx.doi.org/10.1061/(asce)he.1943-5584.0001427.

Line 183: In Fig. 3, It is better to do trend analysis for precipitation or drought. Is the precipitation trend positively increasing or negatively decreasing?

Response: In Fig. 3, we used precipitation anomaly to assess meteorological drought period. The precipitation presented decreasing trends with a slope of -0.76 mm a-2, R2=0.02.

Fig. 1 annual precipitation (P) and it temporal trends.

Line 201: In Table 2, It is necessary to explain how to determine the start time and end time of data period (1975.6-1985.7; 1975.7-1985.8; 1999.6-2012.8; and 1999.7-2013.4) during the drought events.

Response: The start time and end time of data period is assess by the method showed in Fig. 3. The method also showed in method section quantifying drought “(i) By calculating the annual precipitation anomaly, wherein results are smoothed within a 3-year moving window to avoid individual wetter years that can be interspersed between extensive and pronounced dry periods, and (ii) by determining the exact months for the onset and the conclusion of the dry period based on accumulated monthly precipitation anomalies.”.

Different time period could get different results. Line 235-238: The baseflow recovery in this study area generally exhibits a longer time lag. How do the climatic factors and catchment properties in your study area cause the result of longer time lag? It is necessary to explain in more detail. 

Response: This has been improved, and some more details have been added to explain this as “As showed in Table 2, drought severity and intensity were greater (-47.37% and -0.39 in the first period, -83.43% and -0.51% in the second period, respectively), and baseflow recovery correspondingly exhibited a longer time (28 months for the first period, and 30 months for the second period, respectively).”.

Reviewer 2 Report

Extremely well written presentation of an interesting research topic. Methods appear to be appropriate and the analysis sufficiently rigorous. The only suggestion I have is to revise the conclusions to reiterate the importance of this study to ongoing research in this area. Perhaps, some examples of potential applications of methods would also be useful. 

Author Response

Response: The conclusions have been improved to highlight this study in regulating the e-flow to maintain the integrity of river ecosystem. For example, explanations for linkages between flow regimes, biota, and ecosystem processes will help define e-flow while also helping to protect and restore aquatic ecosystems in specific catchment, which has been address in the discussion section.

Reviewer 3 Report

Review of manuscript titled, “Lag in hydrological recovery following extreme meteorological drought events: Implications for ecological water requirements” by Q. Liu et al.

Overview

This manuscript describes the use of monthly anomalies in precipitation, streamflow, and baseflow to characterize periods of drought in the Taoer River, China, above the Taonan station. Three metrics of environmental flow were calculated during each of four flow regimes. The study results indicated that hydrologic droughts were of greater duration and intensity than meteorological droughts. 

Comments

Figure 1 is difficult to read, please enlarged it with higher resolution.

Section 2.2 Methods: Please indicate the source of precipitation and flow data used (with URL or citation).

Line 122 and 123: Equation 2 shows Qi which is not defined, and the text defines Qq which is not in the equation. Please ensure that they are consistent.

Line 132: Please define what ABIT stands for and include a citation.

Line 145: Please define C in equation 7

Line 147: Please define what δ stands for in equation 6 and on line 147.

Line 174: Please present the data pertaining to the isotope analysis and add more discussion of this topic.

Line 191: I believe the four percentages in this line should be negative ( per Table 2).

Line 204: Please define how abrupt changes were identified. Was is using abrupt changes in slope, mean, or other metric?

Line 209, Figure 4: Please clearly identify the “abrupt changes” separating the four periods referred to in the text.

Line 218, Table 3: Please explain how Ewb, Ewm, and Ews are calculated and include citations.

Line 246: The linkage between drought and environmental flow could be more fully developed. The discussion in Section 4.2 is somewhat confusing, particularly when the conversation transitioned to groundwater.

Author Response

Review of manuscript titled, “Lag in hydrological recovery following extreme meteorological drought events: Implications for ecological water requirements” by Q. Liu et al. Overview This manuscript describes the use of monthly anomalies in precipitation, streamflow, and baseflow to characterize periods of drought in the Taoer River, China, above the Taonan station. Three metrics of environmental flow were calculated during each of four flow regimes. The study results indicated that hydrologic droughts were of greater duration and intensity than meteorological droughts. Comments Figure 1 is difficult to read, please enlarged it with higher resolution. Response: Figure 1 has been improved. Section 2.2 Methods: Please indicate the source of precipitation and flow data used (with URL or citation). Response: The source of precipitation and flow data were added in the improved manuscripts. Line 122 and 123: Equation 2 shows Qi which is not defined, and the text defines Qq which is not in the equation. Please ensure that they are consistent. Response: Corrected! Line 132: Please define what ABIT stands for and include a citation. Response: Improved! ABIT has been explained in line ?, and a citation has been added in line ?. Line 145: Please define C in equation 7 Response: The mistake has been corrected. The equation has been revised as: X=((δ_n-δ_2 ))⁄((δ_1-δ_2 ) ). Line 147: Please define what δ stands for in equation 6 and on line 147. Response: Isotopic compositions (δ) represent deviations in per mil (‰) from Vienna Standard Mean Ocean water (VSMOW) such as δ_sample=[(R_sample⁄R_VSMOW )-1]×〖10〗^3, where R is the 18O/16O or 2H/1H ratio in sample and VSMOW. This has been added in the revised manuscript. Line 174: Please present the data pertaining to the isotope analysis and add more discussion of this topic. Response: The data pertaining to isotope analysis has been added in the data section. The sample site also showed in the Fig. 1. Line 191: I believe the four percentages in this line should be negative ( per Table 2). Response: Corrected! Line 204: Please define how abrupt changes were identified. Was is using abrupt changes in slope, mean, or other metric? Response: The abrupt changes method has been added in the method section. Line 209, Figure 4: Please clearly identify the “abrupt changes” separating the four periods referred to in the text. Response: The Fig. 4 has been improved, detailed information have been added. Line 218, Table 3: Please explain how Ewb, Ewm, and Ews are calculated and include citations. Response: The Tennant method also has been added in the method section. Line 246: The linkage between drought and environmental flow could be more fully developed. The discussion in Section 4.2 is somewhat confusing, particularly when the conversation transitioned to groundwater. Response: Absolutely agreed with you. Environmental flow (e-flow) used to link the flow regimes with aquatic habitat. E-flow which is rooted in the natural flow regime paradigm has been assessed and applied to restore a range of flow regimes. As shown in Fig. 3 and Table 1, we hardly defined periods outlining natural flow regimes, and used this information to link flow regimes with ecological structure and function. Natural flow regimes depended on times period used to define the natural flow. In this context, we need to broad static states to dynamic states, as showed in the discussion section “the challenges in determining e-flow involve transitions from relatively static, regime-based hydro-ecological characteristics to those that are more dynamic as well as time-varying flow characteristics that can be linked mechanistically to a wide range of ecological performance metrics applicable over a range of spatial and temporal scales”. As the groundwater, it can be used to reflect the water storage state in the basin scale. Alteration of groundwater should influence the variation of hydrological regimes in baseflow and streamflow. As your comments, some mistakes has been corrected, e.g., (As shown in Fig. 3 and Table 1, we comprehensively defined periods outlining natural flow regimes, and used this information to link flow regimes with ecological structure and function) has been improved as “As shown in Fig. 3 and Table 1, we comprehensively defined periods outlining natural flow regimes, and used this information to link flow regimes with ecological structure and function”.

Round 2

Reviewer 1 Report

Accept in this revised form.

Author Response

The manuscripts has been imroved in methods sections, and some errors also were corrected.

If you have any other comments, queries, or suggestions, we will be happy to answer them. Please feel free to contact us.

Reviewer 3 Report

Thank you for your revision. I have a few remaining comments. 

Line 170 and Figure 4: The M-K test is typically used to determine if a trend is statistically significant or not. It is not clear how the M-K test was used to identify abrupt changes. It is also unclear in Figure 4b whether the arrows are pointing to the dashed line, the solid line, or the intersection of the two. I reread the text several times and cannot determine what UFk and UBk refer to. Please provide more explanation of what Figure 4b is illustrating and how you interpreted abrupt changes. Please also describe the y axis ("u" is not descriptive enough).

Line 193: Second request - please present the results of the stable isotope analysis. Only the result (groundwater=33.1%) is presented on line 194 without any supporting data.

line 275: "we hardly defined periods..." The word "hardly" does not make sense in the context of the sentence. Please seek a more appropriate word to replace "hardly".

Line 302: Please correct grammar in the sentence beginning "All of these..." (perhaps delete the word "be"). Also check the other recently revised or added sentences for minor grammar errors (e.g., line 264 is missing a %).

Author Response

Dear Professor,

We greatly appreciate your comments for providing us the opportunity to improve the quality of our manuscript. We consider your comments and implemented all of them. The comments have been addressed in the response letter. We hope that our responses and the revised (and substantially improved) manuscript will answer all your queries and that it is now suitable for publication in Water. However, if you have any other comments, queries, or suggestions, we will be happy to answer them. Please feel free to contact us.

Yours!

Qiang LIU
